# Phosphorus natural background estimation in the Scheldt river using tidal marsh sediment cores

Florian Lauryssen [1], Philippe Crombé [2], Tom Maris [3], Elliot Van Maldegem [2], Marijn Van de Broek [4], Stijn Temmerman [3], Erik Smolders [1]

[1]Division of Soil and Water Management, Department of Earth and Environmental Sciences, KU Leuven, Kasteelpark Arenberg 20 bus 2459, 3001 Leuven, Belgium

[2]Department of Archaeology, Ghent University, Sint-Pietersnieuwstraat 35, 9000, Ghent, Belgium

[3]University of Antwerp, Ecosystem Management Research Group, Campus Drie Eiken, D.C.120, Universiteitsplein 1, 2610 Wilrijk , Belgium

[4]Sustainable Agroecosystems group, Department of Environmental Systems Science, Swiss Federal Institute of Technology, ETH Zürich, Zürich, Switzerland

*Correspondence to*: Florian Lauryssen (florian.lauryssen@kuleuven.be)

**Abstract.** Elevated phosphate ($PO_4$) concentrations can harm the ecological status in water by eutrophication. In the majority of surface waters in lowland regions such as Flanders (Belgium), the local $PO_4$ levels exceed the limits defined by environmental policy and fail to decrease, despite decreasing total phosphorus (P) emissions. In order to underpin the definition of currents limits, this study was set up to identify the pre-industrial background $PO_4$ concentration in surface water of the Scheldt river, a tidal river in Flanders. We used the sedimentary records preserved in tidal marsh sediment cores as an archive for reconstructing historical changes in surface water $PO_4$. For sediment samples at sequential depths below the marsh surface, we dated the time of sediment deposition and analysed the extractable sediment-P. The resulting time series of sediment-P was linked to the time series of measured surface water-$PO_4$ concentrations (data 1967-present). By combining those datasets, the sorption characteristics of the sediment could be described using a Langmuir type sorption model. The model requires careful consideration of P migration and correlates DPS with $PO_4$ to reconstruct historical concentrations. The calibrated sorption model allowed us to estimate a pre-industrial background surface water $PO_4$ levels, based on deeper sediment-P that stabilised at concentrations smaller than the modern. In three out of the four cores, the sediment-P peaked around 1980, coinciding with the surface water $PO_4$. The estimated pre-industrial (~1800) background $PO_4$-concentration in the Scheldt river water was 62 [57; 66 (95%CI)] µg $PO_4$-P L$^{-1}$. That concentration exceeds the previously estimated natural background values in Flanders (15-35 µg TP L$^{-1}$) and is about half of the prevailing limit in the Scheldt river (120 µg $PO_4$-P L$^{-1}$). In the 1930s, river water concentrations were estimated at 140 [128; 148] µg $PO_4$-P L$^{-1}$, already exceeding the current limit. The method developed here proved useful for reconstructing historical, background $PO_4$ concentrations of a lowland tidal river. A similar approach can apply to other lowland tidal rivers to provide a scientific basis for local, catchment specific $PO_4$ backgrounds.

## 1 Introduction

Elevated phosphorus (P) concentrations in surface waters is a global problem (Azevedo et al., 2015; Dodds and Smith, 2016; Elser et al., 2007). Eutrophication by excess nutrients, including P and nitrogen (N), can lead to hypoxia, acidification, and harmful algal blooms (Azevedo et al., 2015; Correll, 1998; Watson et al., 2018). Therefore, limiting P concentrations in the surface water is crucial to ensuring a good ecological status, human health and well-being. Downstream systems are at higher risk for nutrient-stimulated eutrophication (Watson et al., 2018). As a result, eutrophication of lowland rivers is on the international agenda, and P is considered the limiting nutrient (Jarvie et al., 2006; Mainstone and Parr, 2002; Reynolds, 2000). Like Flanders (North of Belgium), many lowland regions (The Netherlands, Germany) do not achieve good water quality mainly due to the excess of nutrients (Bitschofsky and Nausch, 2019; Huet, 1990; Van Der Molen et al., 1998; Van Puijenbroek et al., 2014; Rönspieß et al., 2020; Schulz and Herzog, 2004). However, the natural background concentrations of lowland river $PO_4$-P may be higher than those of upland rivers because of biogeochemical processes typical for such waters. For example, diatom assemblages revealed natural eutrophic conditions in The Spree river in Germany, with total P (TP) concentrations of 80 µg $L^{-1}$, compared to recent data of 120 µg TP $L^{-1}$ (Schönfelder and Steinberg, 2004; Zak et al., 2006). Therefore, estimating the background is essential for developing nutrient limits, as it provides a baseline for water quality assessments. It is crucial to ensure the P form when interpreting surface water P concentrations. Phosphorus is either present in solution (dissolved P) or associated with the suspended matter. This study focused on dissolved orthophosphate ($PO_4$), almost identical to the reactive P determined by a colour reaction. Other P forms present in surface water include organic P fractions, and P adsorbed to mineral colloids. Total P refers to all P forms together. The environmental limits are either expressed as reactive P (equated to $PO_4$-P limits), as TP limits or both.

Since 2000, the European Union has regulated surface water quality with the Water Framework Directive (WFD). The WFD does not prescribe limits for P in rivers and lakes but provides a framework for local regulations. The local maximum P concentrations identify pristine environments with minimal anthropogenic disturbance, i.e. the natural background (EU-Parliament, 2000). However, the definition of the natural background has been subject to debate for many river basins (Matschullat et al., 2000; van Raaphorst et al., 2000). Here we define, the natural background as those concentrations found in the environment without any human activity, reflecting only natural geochemical processes (Laane, 1992; Reimann and Garrett, 2005). This definition implies that concentrations have to be estimated before human activity, which is not always feasible. Therefore, a pre-industrial background can be defined instead, inferred from samples dating before the industrial revolution (Reimann and Garrett, 2005). The pre-industrial background is logically affected by anthropogenic processes. For example, in Belgium, the industrial revolution started around 1800 with three million inhabitants (Vanhaute, 2003). Before that, large scale agriculture dates back to the middle ages and the Roman period. However, the most substantial increase in nutrient emissions occurred after the 1950s due to sewer infrastructure, mineral fertilisers and P-loaded detergents (Billen et al., 2005; van Raaphorst et al., 2000).

For the assessment of historical river water quality, sediment analysis is valuable. In surface waters, sediments can both serve as a sink or a source of $PO_4$, depending on the sediment surface chemistry and water concentrations (Froelich, 1988; House and Denison, 1998; Simpson et al., 2021; van der Zee et al., 2007). For example, P storage on fine bed sediments can amount to 60% of the total P export of a catchment nutrient budget (Ballantine et al., 2009; Svendsen and Kronvang, 1993). The essential processes for $PO_4$ are adsorption and desorption from Fe oxy-hydroxides, present in the suspended matter or bed sediments (Froelich, 1988; van Raaphorst and Kloosterhuis, 1994; van der Zee et al., 2007). Ferric iron- (Fe(III)) and aluminium-oxyhydroxides have a high affinity for $PO_4$-anions and limit the $PO_4$ in solution (Borggaard, 1990; Holtan et al., 1988). As a result, the surface water $PO_4$ concentration depends on the sorption capacity of Fe oxy-hydroxides, which decreases with increasing pH and salinity (van der Zee et al., 2007). However, anoxic conditions lead to the reductive dissolution of those Fe(III) minerals. As a result, the associated P is released to the overlying water when the sediment is strongly reduced (Baken et al., 2015). It is now well established that such reducing conditions explain the typical summer peaks in $PO_4$ and that regional differences in sediment Fe concentrations explain regional differences in surface water $PO_4$ concentrations in Flanders (Smolders et al., 2017).

Several authors in Belgium and the Netherlands described the relation between soil characteristics and pore water $PO_4$ concentrations. They used soil analysis to identify agricultural areas sensitive to $PO_4$ leaching as the soil P content showed a good correlation with pore water P concentrations (Breeuwsma et al., 1995; Lookman et al., 1995; Schoumans and Chardon, 2015; Schoumans and Groenendijk, 2000; van der Zee, 1988). Unfortunately, a similar relation between sediments deposited by rivers and surface waters has not yet been described. However, sediments-P can likely predict surface water $PO_4$ concentrations because the P adsorption characteristics of sediments have been related to surface water $PO_4$ concentrations (Wang et al., 2009; Zhou et al., 2005). In addition, streams sediment can buffer $PO_4$ concentrations, and by sediment analysis, we can identify if stream sediment acts as a source or a sink for dissolved P (Jarvie et al., 2005; McDowell, 2015). Thus, the existing literature shows that analysis of sediment P content could be related to surface water P concentrations, similar to what has been observed in soils and pore water.

In addition, sediment P-analysis has shown to be relevant for the long term reconstruction of P in the environment. For example, Boyle et al. (2015) used P profiles from lake sediments in the UK to infer the historical evolution in population density over 10.000 years in catchments. Similarly, banded iron formations in deep oceanic waters allowed to infer oceanic P concentrations over two billion years ago (Bjerrum and Canfield, 2002). The sediments deposited by rivers or lakes react with surface water $PO_4$ and can be deposited in regularly flooded areas. Thus, those sediments can serve as an archive for reconstructing historical P emissions trends and provide useful information on historical $PO_4$ concentrations in adjacent water bodies (Birch et al., 2008).

In lowland rivers with tidal influence, like the Scheldt, vegetated tidal marshes develop along the river banks. Tidal marshes directly adjacent to tidal rivers are regularly flooded during high tides, so river sediments and associated elements like P are deposited on these densely vegetated marshes (Friedrichs and Perry, 2001; De Swart and Zimmerman, 2009; Temmerman et

al., 2004a). As a result, the elevation of tidal marshes increases over time due to their net accumulation of sediments (Temmerman et al., 2003a). Therefore, researchers have used tidal marshes as sediment archives of deposited substances other than P, such as organic carbon (Van de Broek et al., 2019) and silicon (Struyf et al., 2007). However, it remains to be investigated to what extent P concentrations measured in tidal marsh sediment archives can be used to reconstruct historical changes in $PO_4$ concentrations in the adjacent estuary.


In Flanders, environmental pressure is high, and eutrophication affects most water bodies (European Commission, 2019). The average $PO_4$ concentration in Flemish waterways stabilised at 290 µg $PO_4$-P $L^{-1}$, well above the limits varying between 70-140 µg $PO_4$-P $L^{-1}$ for different river types (Smolders et al., 2017; VMM, 2018). However, despite the current net-zero P-balance in agricultural soils, the situation did not improve since 2004 (MIRA, 2017). Therefore, the question arises when or

even if these limits can be achieved. Surprisingly, the prevailing limits in Flanders lack a scientific basis and are not adapted to the local situation (Fien Amery, 2015; Schneiders, 2007). Flanders is densely populated, and it was considered impossible to locate pristine or reference lakes in the current environment to develop nutrient limits. Instead, natural background TP concentrations for Flanders have been inferred from reference lakes sampled in Central and Baltic states in Europe identified for the WFD (Cardoso et al., 2007). Based on that study, background TP concentrations in lakes of Flanders were set at 15-35

µg TP $L^{-1}$, selected for lakes with representative depth and alkalinity. Until now, no TP or $PO_4$ natural background has been established for rivers in Europe (Salminen et al., 2005).

This study tested and evaluated a methodology to estimate the pre-industrial background river water $PO_4$ concentrations based on the analysis of tidal marsh sediment, which has been deposited on the banks of Flander's largest tidal river over multiple

centuries. Using our results, we could provide the first estimate of pre-industrial $PO_4$ levels in a large lowland river, the Scheldt. First, we described the tidal marsh sediment sorption characteristics by linking the P concentration in dated tidal marsh sediments to historical measurements of $PO_4$ in the Scheldt river water. Those sorption characteristics allowed us to estimate river water $PO_4$ concentrations based on analysis of sediments deposited in the 1800s before industrialisation. The underlying assumption is that sediment-P remains immobile and that the sediments depth profile reflects the historical trend of $PO_4$ in the

Scheldt river. Accordingly, we argue that the sediment P-composition in deeper sediment layers of tidal marshes provides an estimate of the historic $PO_4$ concentration of the adjacent river. A database containing measurements of the $PO_4$ concentration in the Scheldt river's surface water (1967-current) verified this assumption. We hypothesise that the previously estimated natural background of this major lowland river is larger than estimated earlier for lakes (15-35 µg TP $L^{-1}$).

## 2 Materials and methods

### 2.1 Study area

Freshwater tidal marshes were sampled at four locations along the Scheldt river (Fig. 1, Table S1). The Scheldt estuary is located in northern Belgium and the south-western Netherlands and flows into the North Sea. The river basin of the Scheldt covers a large part of Flanders (71%) and the adjacent region of Northern France; the total catchment area is approximately 22.000 km$^2$. The population living in the river basin is about 10 million (Meire et al., 2005). The tidal wave extends from the mouth (Vlissingen) to 160 km upstream near Ghent, where sluices stop the tidal wave. The estuary's freshwater tidal zone reaches from Ghent to Rupelmonde (Fig. 1). This research focused on freshwater tidal marshes, i.e. located in this freshwater tidal zone of the estuary. Brackish waters experience the mixing of seawater, making it difficult to distinguish the anthropogenic sources from seawater influence. Furthermore, saltwater in the North Sea has PO$_4$ concentrations about a factor ten lower than fresh water in the Scheldt river (Burson et al., 2016). This research was focused on freshwater, lowland river systems and the human influence on the P concentrations, saltwater environments were beyond the scope of this study.

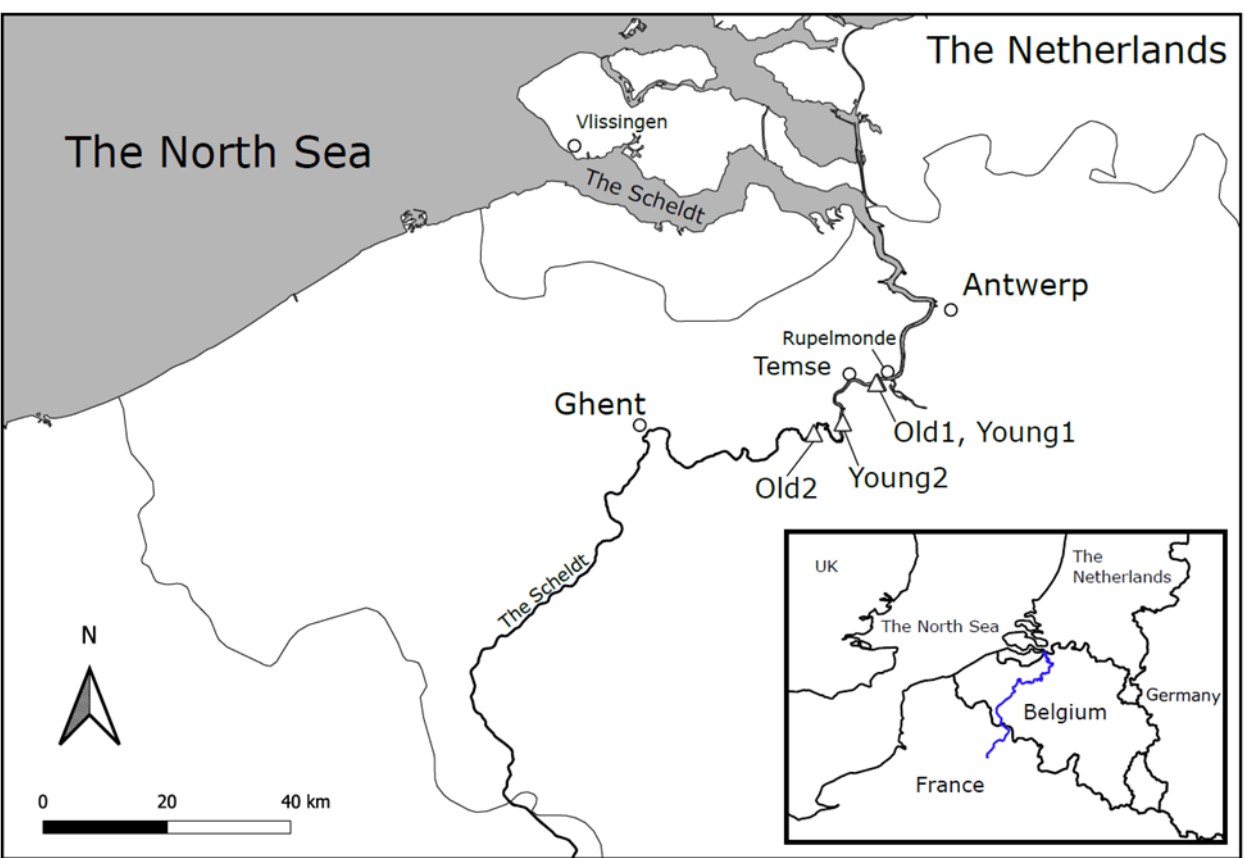

**Figure 1: Map of the Scheldt Estuary, triangles indicate the locations of the sampled tidal marshes, Old1 and Young1 were only 250 m apart, and on the scale of the map, they overlap**

Sediment accreting in tidal marshes originates from the deposition of riverine suspended matter, including inorganic mineral sediment and organic matter (Callaway et al., 1996). We discriminate between old and young tidal marshes, hereafter referred to as marshes. Old marshes have a higher elevation compared to young marshes. As a general mechanism, young marsh surfaces accumulate sediments quickly and increase their elevation asymptotically up to an equilibrium level around the mean high water level (MHWL)(Pethick, 1981; Temmerman et al., 2003a). Temmerman et al. (2003a) defined an old marsh as

visible on topographic maps of Ferraris (1774 - 1777), so it was formed before the 19th century (Temmerman et al., 2003a). Young marshes in the Scheldt estuary were formed more recently, by the natural establishment of pioneer marsh vegetation on formerly bare tidal mudflats, generally after 1944. During the last decades, the young marshes had a surface elevation below MHWL. As a result, young marshes experienced more frequent inundations and therefore had larger sediment accretion rates than old marshes. For example, between 1931 and 1951, young marshes accumulated sediments at rates of 1.6 to 3.2 cm yr$^{-1}$,

and during the subsequent period 1955-2002, accumulation was slower at 0.4-1.8 cm yr$^{-1}$. In contrast to the young marshes, the elevation of old marshes is at any time was very close to the yearly MHWL increase rate of 0.3 to 0.6 cm yr$^{-1}$ in the Western Scheldt (Temmerman et al., 2003a).

     This study analysed depth profiles of sediment cores originating from tidal marshes along the freshwater Scheldt river. The analysis contained two old and two young marshes, referred to as Old1, Old2, Young1 and Young2 (locations indicated in Fig.

1). The coordinates of sampling locations can be found in (Van de Broek et al., 2018; Van De Broek et al., 2016) and supplementary information (SI.I). Marshes Old1 and Young1 originated from the tidal marsh named the Notelaer, Old 2 from Grembergen and Young2 from Mariekerke. In total, we analysed eight cores; three replicate cores for both sites Old1 and Young 1 and one core each for Old2 and Young2.

## 2.2 PO₄ concentration in surface waters

The IMIS (Flanders Marine Institute) provided surface water phosphate (PO₄) data measured colourimetrically on a filtered water sample and total phosphorus (TP) by acid digestion and a segmented flow analyser. Data of PO₄ concentrations in Scheldt river were available from 1967 up to 2019 compiled by the OMES-monitoring, who did additional quality controls on the data (ECOBE - UA; The Flemish Waterway, 2019). The PO₄ data originated from different sources described in the supplementary information (SI.V) (De Pauw, 2007; ECOBE - UAntwerpen, 2007; Institute for Hygiene en Epidemiology

(IHE), 2007; *OMES: Monitoring physical-chemical water quality in the Zeeschelde*, 2016; Van Meel, 1958). The open-source software R (R Core Team, 2020) was used to compile all available datasets for PO₄ closest to the study sites (Temse) and to calculate annual means by averaging all observations within each year for which data was available. The annual means of PO₄ were used to visualise the evolution of PO₄ in the Scheldt river (Fig. 2). Because P emissions mainly originate from point sources due to domestic loading, the surface water P concentration increased between 1950 and 1975 related to the rise in the

number of households connected to sewer systems, without appropriate wastewater treatment being in place (Billen et al., 2005). Since 1985, wastewater treatment has significantly improved the situation (Billen et al., 2005).

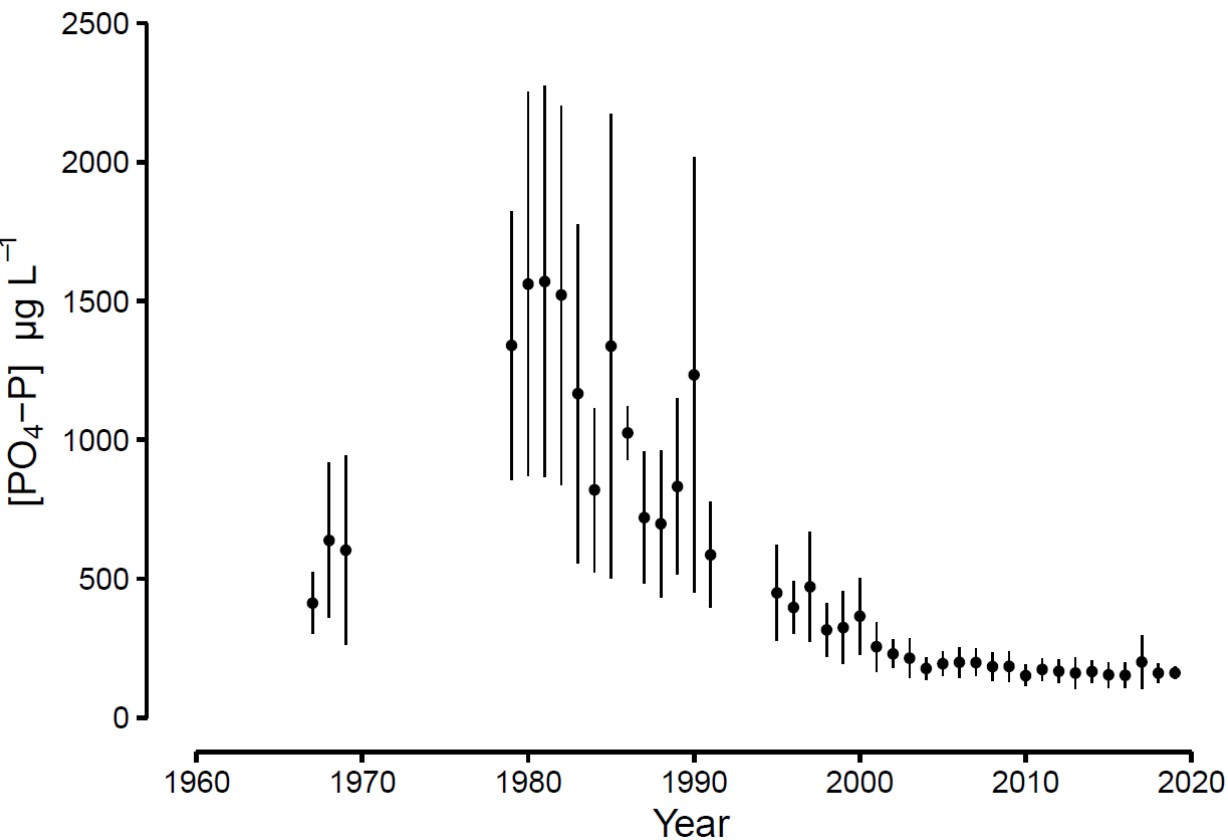

**Figure 2. Concentrations of phosphate (PO4-P) in the Scheldt River at Temse, annual means and standard deviation (error bar)**
**around the annual mean. Samples were taken in Temse close to tidal marsh sites. (data sources: (ECOBE - UAntwerpen, 2007; Institute voor Hygiëne en Epidemiologie (IHE), 2007; Van Meel, 1958; Nv, 2016; De Pauw, 2007)**

## 2.3 Sediment sampling

The sediment samples used here had been collected during a previous study about carbon sequestration in tidal marsh sediments in the Scheldt estuary (Van de Broek et al., 2018; Van De Broek et al., 2016). Collection of undisturbed sediment profiles on
the tidal marshes took place between July and September 2016 (Old1, Young1, Old2, Young2; Fig. 1). Undisturbed sediment cores were taken using a gauge auger (0.06 m diameter) at each sampling location. In the field, the cores were divided into subsamples with a 0.03 m interval. The sediment samples were dried at a maximum temperature of 50 °C for 48 hours, crushed and sieved to a <2 mm grain size. Macroscopic vegetation residues were removed manually using tweezers (Van de Broek et al., 2018). Bulk density, grain size distribution and organic carbon (OC) content were analysed by Van de Broek *et al.* (2018).
We refer to Van de Broek et al. (2016, 2018) for further information about sample collection and processing.

## 2.4 Sediment analysis

The dried sediment samples were analysed for oxalate-extractable P, Fe, Al and Mn ($P_{ox}$, $Fe_{ox}$, $Al_{ox}$, $Mn_{ox}$; Schwertmann, 1964). The preparation of extraction solution and dilutions were made with ultrapure water (Milli-Q®), and all glassware used was acid-soaked overnight in a 1% HCl acid bath to prevent P contamination. That acid oxalate extractant, a mixture of ammonium oxalate (0.2 M) and oxalic acid (0.2 M) at pH = 3, targets poorly crystalline oxyhydroxides of Fe, Al and Mn and the associated P (Schwertmann, 1964). Poorly crystalline oxyhydroxides are the most reactive due to their large specific surface area (Hiemstra et al., 2010). The extraction was done with 1 g of dry sediment in 50 ml extraction solution over two hours in an end-over-end shaker at 20°C (26 rpm). The suspension was filtered through a 0.45 µm membrane filter (CHROMAFIL ® Xtra PET - 45/25). Analytical blanks, internal reference samples, and duplicate samples were included in every batch to ensure the analysis's quality, purity, and reproducibility. The extract was diluted 20 times and measured by inductively coupled plasma optical emission spectrometry (ICP-OES). The degree of P-saturation (DPS; %) was calculated as in Eq. (1). The DPS represents the ratio of extractable ($P_{ox}$) to the P sorption capacity of the sediment. This P sorption capacity is estimated as half of the sum of oxalate $Fe_{ox}$ and $Al_{ox}$, because not all the Fe and Al in a soil is available for P sorption

$$DPS = \frac{P_{ox}}{0.5(Fe_{ox} + Al_{ox})} 100\%, \qquad (1)$$

with $Fe_{ox}$, $Al_{ox}$ and $P_{ox}$ in molar units.

The DPS is expressed in percentage and can be interpreted as the ratio of sorption sites on the sediment occupied by P. Previous research used the DPS to identify agricultural areas sensitive to phosphate leaching and showed a good correlation with pore water P concentrations (Breeuwsma et al., 1995; Lookman et al., 1995; Schoumans and Chardon, 2015; Schoumans and Groenendijk, 2000; van der Zee, 1988). The DPS was verified for porewater-soil systems and was developed by (van der Zee et al., 1990). The factor 0.5 is an empirical value based on pore water measurements of non-calcareous sandy soils and is considered the sorption capacity of the soil. Lexmond et al. (1982) illustrated that the maximal sorbed P was about half the pool available after a long-term precipitation experiment. The parameter α primarily affects the maximum sorption capacity. So they set α at 0.5. However, even among soils, this parameter varied between 0.3 and 0.6 (Lexmond et al., 1982). We are interested in low background concentrations for this research, so maximal sorption, occurring at high $PO_4$ concentrations, is less relevant.

## 2.5 Age-depth model

The sediment analysis and the surface water $PO_4$ data had to be linked by a corresponding date and location to fit a sorption model. Therefore, an age-depth model was used to calculate the time since deposition of each sediment sample. Temmerman et al. (2004b, 2004a) developed a model (MARSED) to estimate sediment deposition rates and the resulting evolution of the sediment surface elevation in the tidal marshes of the Scheldt estuary. Hence, we could also use that model to determine the time since deposition of sediments throughout the sampled sediment profiles. The MARSED model simulates the tidal supply

of suspended sediments and the settling of these sediments to the marsh surface during tidal inundation cycles integrated over the years. The model was calibrated and validated against measured data on sediment deposition rates on the Scheldt estuary tidal marshes from 1945 until 2002 (Temmerman et al. 2003; 2004). The empirical data on sediment deposition rates were
derived from radiometric and paleoenvironmental dating of sediment cores at the same locations sampled for the present study (Temmerman et al., 2004a, 2004b). For our current study, we extrapolated the model simulations of sediment accretion from 2002 until 2016, the sampling date of the sediment cores (Van de Broek et al. 2018). We found that the MARSED model overestimated the observed marsh surface elevation in 2016 (observed by RTK GPS surveying; Van de Broek et al. 2018; Poppelmonde, 2017) by 25 cm for sampling location Old1, 29 cm for Young1, 19 cm for Old2, and 8 cm for Young2. The
MARSED model was initially designed to simulate the overall sediment accretion and surface elevation changes in tidal marshes in response to sea-level rise scenarios, for which those errors were acceptable. However, for the present study, we wanted to estimate the time of sediment deposition throughout the sediment profiles as good as possible. Therefore, the original age-depth relation calculated by MARSED was recalibrated by using observed age-depth points. The observed age-depth points originate from GPS measurements of marsh elevation in 2016 (M. Van de Broek, unpublished data; SI.I) and previously
published radiometric and paleoenvironmental dating (Temmerman et al. 2003; 2004). This rescaling procedure is explained in the Supporting Information (Fig. S1, S2, S3, S4). The observed age-depth points were available starting from 1958 for sampling site Old1, 1947 for Young1, 1963 for Old2 and 1968 for Young2 (Temmerman et al. 2004). An approximate extrapolation procedure was used to estimate the sediment deposition time from depths below the oldest measured age-depth points (mentioned in the previous sentence). This extrapolation procedure was only applied for old marshes, which were
defined as marshes that existed at least since the end of the 18[th] century (Temmerman et al. 2003a; 2004). Two sediment cores originated from such old tidal marshes (Old1 and Old2). We know from observed age-depth points that old marshes reached equilibrium with the MHWL before 1944, and that they built up their elevation after 1944 at a rate comparable to the rate of local MHWL rise (Temmerman et al., 2003a). Here, we assumed that between 1800 and 1944, these old marshes also accreted at a rate comparable to the MHWL rise. Historical tide gauge data of MHWL rise was available from 1901 for site Old1 and
1930 for site Old2 (ScheldeMonitor Team and VNSC, 2020; Temmerman et al., 2003a) and linear regression of the MHWL against time was used to estimate the marsh surface elevation before 1944 (Fig. S6, S7). However, the accuracy of the dating will be lower going further back in time. Such extrapolation to earlier dates is not appropriate for young marshes, as they were only formed after 1950 by pioneer vegetation establishment on formerly bare mudflats (Temmerman et al. 2003a; 2004). Those mudflat sediment profiles do not have continuous sedimentary records as tidal marshes and are likely to be disturbed by erosion
and sedimentation alternations (Belliard et al., 2019). Therefore, the sediment deposition time could not be extrapolated for the young marsh sampling locations before the oldest available measured age-depth points, dating back to 1947 for site Young1 and 1968 for site Young2.

**2.6 Relating surface water PO₄ with sediment P: the sediment-water model**

The age-depth model and linear regression of MHWL provided a deposition year for each sediment sample. Through that age-depth relation, the dataset of water PO₄ between 1967-2016 was linked to the sediment DPS for each core. The resulting dataset contained all available surface water PO₄ readings between 1967 and 2016 closest to the tidal marshes in Temse (n = 1932) and a corresponding DPS value of a sediment sample originating from one sediment core or, when available for Old1 and Young1, a mean of the replicate sediment samples. This dataset allowed to fit a sorption model further termed the sediment-water model. Schoumans and Groenendijk (2000) presented a Langmuir-type sorption model to predict PO₄ concentration leaching from a soil layer based on the DPS Eq. (2).

$$[PO_4] = K^{-1} \frac{DPS}{100-DPS}, \tag{2}$$

With [PO₄] phosphate concentration in (kg L⁻¹), K the sorption constant (L kg⁻¹), DPS (degree of P-saturation; %). This model adequately described P sorption in soil across a wide range of pH values, including the Scheldt river pH (Schoumans and Groenendijk, 2000; Warrinnier et al., 2018). The model relies on surface complexation between PO₄ and Fe-, Al-oxyhydroxides in the sediment, determined by a chemical equilibrium between solid (adsorbed) and dissolved PO₄ phase (Warrinnier et al., 2019). The parameter K of existing soil models has been calibrated for soil - pore water system and the sediment-water parameter (K) is unlikely equal. Therefore, the model was calibrated by fitting parameter K on sediment DPS measurements and recent Scheldt water PO₄ measurements. The parameter K (Eq. 2) was calibrated to the most recent Scheldt water data. As a result, the fitted K-value is adapted to the local geochemistry of tidal marsh sediments and the surface water. We explored 16 different scenarios to fit the sediment-water model Eq. (2). These scenarios illustrate the statistical uncertainty surrounding the estimated PO₄ concentrations. The model was fitted separately for each site sediment core or on the combined replicate cores for Old1 and Young1 (SI.VI). Every sediment sample had between one and three replicates, depending on the depth and the site. We entered either the average value of these replicates or the individual replicate DPS values. One sediment sample covered several deposition years, so multiple PO₄ observations corresponded with each sediment sample. Again, the average of all corresponding PO₄ readings was taken, or all available values were used separately, resulting in 16 models (Table S2). For each of these, the parameter estimation of Eq. (2) was fitted by non-linear regression with JMP Pro (Version 15.1.0. SAS Institute Inc., Cary, NC, 1989-2019). Non-linear least squares regression to the PO₄-DPS data was used to estimate the model parameter (K), yielding the lowest sum of squared errors.

**2.7 Evaluation Model Performance**

The predictions of the sediment-water model were evaluated based on several parameters; the Residual Standard Error (RSE), the Nash Sutcliffe Model Efficiency (E) and by plotting the measured surface water PO₄ against predicted PO₄ between 2007 and 2016 (Table S2; Fig. S10). Additionally, the per cent bias (Pbias) was calculated for data points between 2007 and 2016. The Pbias measures the average tendency of the simulated data to be larger or smaller than their observed counterparts

expressed as a percentage of the observations (Moriasi et al., 1983; Eq. 3). The prediction of recent years is interesting to evaluate the model's performance because of two reasons. First, the most recent surface water $PO_4$-concentrations are relatively low and more representative of background concentrations. Second, the monitoring data have a high temporal resolution, and the age-depth model is more accurate at shallow depths.

$$PBias = \frac{\sum_{i=1}^{n}(Y_i^{obs} - Y_i^{sim})}{\sum_{i=1}^{n} Y_i^{obs}}$$
(3)

## 3 Results

### 3.1 History of surface water $PO_4$ concentrations

The Scheldt $PO_4$-concentrations varied greatly over the past decades, with the peak in surface water $PO_4$-concentrations between 1975 and 1985 (Fig. 2). In Temse, the annual mean concentrations rose from 410 µg $PO_4$-P $L^{-1}$ in 1967 and peaked in 1980 with 1570 µg $PO_4$-P $L^{-1}$. Between 1990 and 2003, a decrease followed the peak and over the last decade, concentrations stabilised between 160 and 200 µg $PO_4$-P $L^{-1}$ in Temse. Current $PO_4$-levels are a factor two lower than in 1967 and almost a factor ten lower than the peak in 1980 (Fig. 2; Table 1).

### 3.2 Sediment cores

The $P_{ox}$ in the sediments ranged between 370 mg P $kg^{-1}$ and 13,000 mg P $kg^{-1}$, while the DPS ranged between 13% and 94% (Table 1). In all soil cores starting at the surface, the DPS and $P_{ox}$ increased with depth and peaked at about 0.5 m below the surface (Fig. S7, Fig. S8). In deeper (>1.0 m) sediment layers, $P_{ox}$ and DPS decreased and stabilised for Old1, Young1 and Young2 (Table 1). Overall, the $P_{ox}$ increased by, on average, a factor of 3.5 between the surface and the maximum concentrations (Fig. S8, Table1). These peak DPS were deposited between 1960 and 1985 in three of the four sediment cores (Fig. 2). Only the core Old2 reached the peak earlier (ca. 1940-1950). Most importantly for this work, DPS for Old1 showed an apparent stabilisation in deeper or older layers, which indicated undisturbed sediment layers (Fig. 3, Fig. S7).

**Table 1:** The sediment oxalate extractable P ($P_{ox}$) and its Degree of Phosphate Saturation (DPS) of the top, bottom, peak sediment layers at four different tidal marsh locations. Top layers are the sediments closest to the surface, peak layers had maximal $P_{ox}$ and DPS, and bottom layers are those sediments sampled at the largest depth. Values of $P_{ox}$ and DPS are means (± standard deviation) of N sediment samples, between top and bottom (cm) depth.

| Location | | N | Top – Bottom (cm) | $P_{ox}$ (mg kg$^{-1}$) | $Fe_{ox}$ (mg kg$^{-1}$) | $Al_{ox}$ (mg kg$^{-1}$) | DPS (%) |
|---|---|---|---|---|---|---|---|
| **Old1** | **Top** | 4 | 0 - 9 | 2300 (± 2400) | 21000 (± 1400) | 1200 (± 130) | 36 (± 3) |
| | **Peak** | 7 | 27 - 57 | 5400 (± 1300) | 24000 (±4700) | 1800 (± 300) | 70 (± 8) |
| | **Bottom** | 8 | 147 - 180 | 540 (± 110) | 8500 (± 750) | 660 (±41) | 20 (± 4) |
| **Young1** | **Top** | 4 | 0 - 9 | 2700 (± 320) | 24000 ± 1200 | 1400 (± 91) | 37 (± 2) |
| | **Peak** | 6 | 27 - 57 | 8500 (± 3200) | 31000 (±7500) | 2000 (± 120) | 85 (± 15) |
| | **Bottom** | 6 | 129 - 144 | 910 (± 440) | 7000 (± 3900) | 650 (±120) | 40 (± 4) |
| **Old2** | **Top** | 3 | 0 - 9 | 2800 (± 90) | 19000 (± 230) | 1700 (±36) | 45 (± 1) |
| | **Peak** | 3 | 54 – 69 | 8000 (± 1600) | 25000 (± 6500) | 2600 (± 360) | 94 (± 2) |
| | **Bottom** | 3 | 132 - 147 | 1700 (± 620) | 11000 (± 3900) | 1700 (± 300) | 43 (± 6) |
| **Young2** | **Top** | 3 | 0 - 9 | 2700 (± 410) | 39000 (± 7800) | 1300 (± 240) | 23 (± 1) |
| | **Peak** | 3 | 48 - 63 | 7000 (± 1200) | 42000 (± 5400) | 1900 (± 57) | 55 (± 7) |
| | **Bottom** | 3 | 144 - 183 | 3200 (± 110) | 34000 (± 770) | 1400 (± 78) | 31 (± 2) |

Within the first meter, $Fe_{ox}$ was stable in the three soil cores (Old1, Young1, Old2) with concentrations around 20,000 mg kg$^{-1}$, except for Young2 for which $Fe_{ox}$ was a factor two larger (Fig. S9). For Young1 and Young2 a decrease in $Fe_{ox}$ concentration occurred at depths > 1 m. For Old1, $Fe_{ox}$ showed a steady decline from 20,000 mg kg$^{-1}$ at the surface to 10,000 mg kg$^{-1}$ at the bottom of the profile (Fig. S9). The $Al_{ox}$ concentrations showed a similar trend as the $P_{ox}$ concentrations, with an initial increase followed by a decrease with depth. The strong correlations of $Al_{ox}$ and $Fe_{ox}$ with $P_{ox}$ ($r_{Al} = 0.73$ and $r_{Fe} = 0.65$) illustrate the positive effect of Fe and Al oxyhydroxides on P sorption.

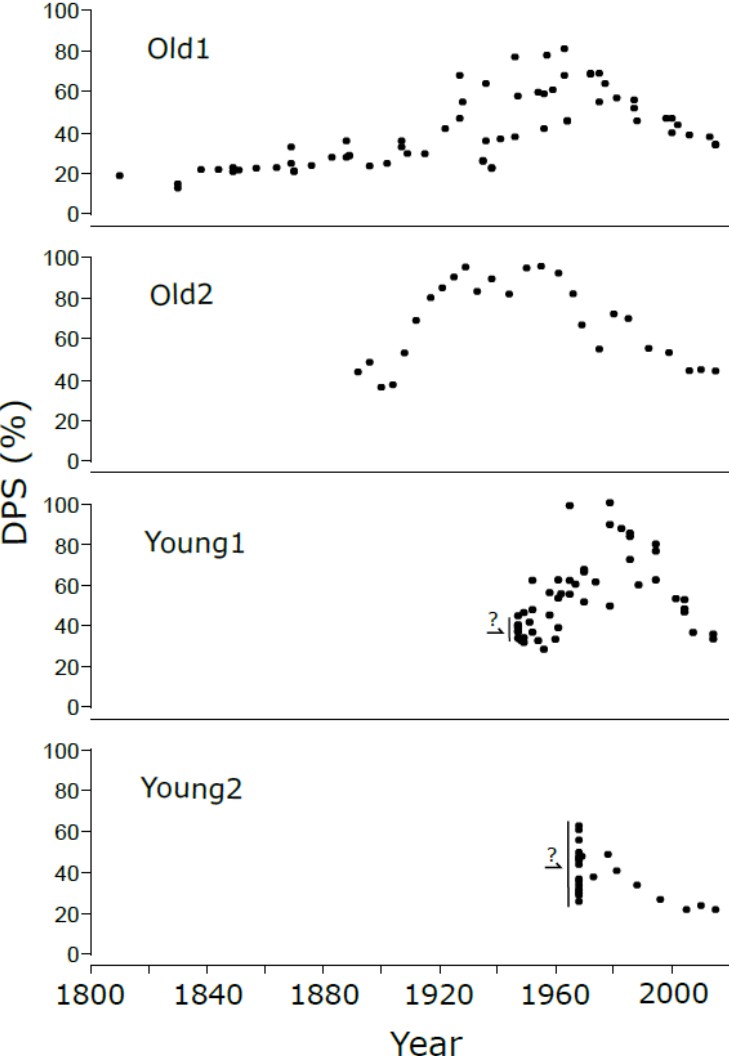

**Figure 3: The Degree of Phosphate Saturation (DPS) timeline based on four tidal marsh sediment sites. Each dot represents a sediment analysis. The year assigned to each sediment analysis was calculated with the age-depth model. Before 1930, no model dates were available. Therefore a linear regression of the MHWL was used to extrapolate the dates for the old marshes. Dates before 1930 are increasingly uncertain going further back in time. For young marshes, such extrapolation was not possible. The points before the formation of the marshes are indicated with a question mark.**

## 3.3 Sediment core selection

Under the assumption that $PO_4$ does not migrate, the tidal marsh sediment cores can provide an archive for river water $PO_4$.
However, that assumption may be most violated at two locations, Old2 and Young2. Considering P-migration, evaluating the distance from a creek within the tidal marsh is crucial. That distance is essential for two reasons. First, within 10 to 20 m of the creeks, the groundwater table fluctuates largely with the tides (Van Putte et al., 2020), which can induce vertical P-migration. Secondly, sediment accretion is more difficult to predict at closer distances to the creeks and can affect the age-depth relation (Temmerman et al., 2003b). The nearest creek for the tidal marshes is within 21 m for Old1, 56 m for Young1,
35 m for Young2 and 5 m for Old2.

Furthermore, the profile of Old2 had a peak of $P_{ox}$ at an earlier date (1950) than was expected from surface water data (1980), indicating P-migration (Fig. 2). Consequently, Old2 was not taken up for interpretation of the relation between DPS and $PO_4$. For core Young2, deeper sediment layers had a larger DPS than the surface layers (Table 2). Additionally, the age estimation of sediments older than 1968 was not possible due to this tidal marsh's young age, which hampers the interpretation of DPS
values from deeper layers as representative for background values. Furthermore, $Fe_{ox}$ concentrations were a factor two larger than the other cores (Fig. S9) and a factor two larger than the average sediment Fe-concentration of the Upper-Scheldt basin (VMM, 2019). Finally, the local enrichment in iron lowers the DPS values and makes the core less representative of the average situation in the Scheldt. These observations made Young2 inappropriate to fit the relation between DPS and $PO_4$.

The two remaining soil cores, Old1 and Young1, originate from the same tidal marsh area named "The Notelaer", located near
the city of Temse (Fig. 1) and has been the topic of multiple studies on sediment accretion (Temmerman et al., 2004b, 2003a) and soil OC stocks (Van de Broek et al., 2018; Van De Broek et al., 2016). The sediment profiles of both sites Old1 and Young1 rise and fall in DPS comparable to dynamics in surface water $PO_4$-concentrations (Fig. 2, Fig. 3). In deeper sediment layers, DPS and $P_{ox}$ stabilise below levels of recent deposits (Fig. S7, S8). The time series of Old1 displayed a DPS peak around 1960, indicating a shift of 20 years (Figure 2). However, the core Old1 was taken up for the prediction of the model
fitting. That core is essential to predict the background because the core dates back to 1800 at the deepest levels. Furthermore, the DPS concentrations stabilised before 1920, indicating that P has not migrated to these depths, making it suitable for background prediction. These observations suggested a well-preserved $P_{ox}$ and DPS profile, essential for the DPS-$PO_4$ relation. Therefore, Old1 and Young1 are considered the best profiles for applying the sediment-water model and interpretation of background concentrations.

Table 2: The predicted concentrations of phosphate (PO₄-P µgL⁻¹) in the Scheldt river based on the Degree of Phosphate Saturation (DPS) in the sediment layers of marsh Old1, dating back to 1800 (pre-industrial), where DPS values stabilised with depth at 20% and the predicted concentration dated to 1930 where DPS stabilised at 36%. The Pbias is the mean difference of the simulated data and the observed between 2007 and 2016, expressed as a percentage of the observed data. Conversion of DPS to river phosphate concentration based on the association of DPS with PO₄-P (Eq. 2) calibrated to data 1967-2016, thereby using different calibrations for sediment-water models; the details of models are in Table S2. Model 3b (in bold), is proposed as the most accurate one (see text).

| Model # | K (L kg⁻¹) [95% CI] | pre-industrial background | | Pbias |
| --- | --- | --- | --- | --- |
| | | ~1800 µg PO₄-P L⁻¹ [95% CI] | ~1930 µg PO₄-P L⁻¹ [95% CI] | (2007 -2016) |
| 1b | $2.1\times10^6$ [$2.0\times10^6$; $2.3\times10^6$] | 120 [110; 130] | 270 [245; 281] | 62 |
| 2b | $4.9\times10^6$ [$4.6\times10^6$; $5.2\times10^6$] | 51 [49; 54] | 120 [109 ; 122] | -28 |
| **3b** | **$4.1\times10^6$** **[$3.8\times10^6$; $4.4\times10^6$]** | **62** **[57; 66]** | **140** **[128; 148]** | -15 |

### 3.4 Sediment-water model fit

The sediment-water model Eq. (2) was fitted on DPS-PO₄ data from the different sediment cores (Table S2). Two observations were omitted because the DPS values were too large (0.99 – 1.02) and produced artefacts in the results. The Nash-Sutcliffe model efficiency (E) ranged between 0.04 and 0.85 depending on the input data (Table S2; Nash and Sutcliffe, 1970). The sediment-water model was fitted on each core's data separately and for the combination of the data from Old1 and Young1, as they came from the same tidal marsh location. The models fitted on data from sites Old2 and Young2 were not considered as migration likely affected those cores (crf. section 3.3).

The models fitted on an average DPS (across replicates) associated with individual PO₄ readings were considered most suitable (Models 1b, 2b, 3b; Table S2). A single sediment sample analysis represents an average P signal over the sediment's deposition period. However, due to variation in the marsh surface elevation, the age-depth relation can vary slightly. By taking an average DPS from replicate cores, we reduced the variation in the independent variable to predict PO₄ in water. Furthermore, in most models, the prediction error increased by relating individual rather than mean DPS values with individual PO₄ measurements (Table S2). Models using unique DPS associated with single PO₄-data duplicated or even triplicated the PO₄-data, artificially creating more degrees of freedom (Model 1c, 2c, 3c; Table S2). Using mean PO₄-values artificially reduced the degrees of freedom, compromising the model predictions by increasing RSE and widening confidence intervals (Models 1a, 2a, 3a; Table S2). The fitted parameter K (L kg⁻¹) ranged between $1.0\times10^6$ and $5.4\times10^6$ for the different input datasets, with the 95%

confidence intervals ranging between $0.8\times10^6$ and $7.2\times10^6$. The variation of parameter K for the various input datasets was larger than the individual confidence limits variation (Table S2). Thus, the uncertainty was more pronounced due to the variability in sediment samples than due to the model fit.

## 3.5 Model performance

The sediment-water model performance was evaluated with several parameters (RSE, Pbias, and E), and by comparing the actual by predicted $PO_4$ concentrations over the last decade, those $PO_4$ concentrations are more comparable to the background (Fig. S10; Table 2; mean Temse [2007-2016] = 170 µg $PO_4$-P$L^{-1}$). Model 3b was considered the most suitable for predicting background concentrations. The Pbias was the lowest for recent observations for model 3b. The average tendency of simulated data compared to the observations was only -14.9 %, which is within the acceptable range of ±25% (Moriasi et al., 1983). Model 2b had an underestimation of more than 28%, and model 1b overestimated recent observations by more than 60%, which is unwanted for calculating the background and were both considered unsuitable (Table 2). The actual by predicted plots illustrate a similar message (Fig. S10). Based on these observations, model 3b was considered the best model, although the residual standard error (RSE) was lower for model 2b (Table S2). The selected model 3b successfully reconstructs the rise and fall in surface water $PO_4$-concentrations based on the sediment characteristic DPS (Fig. 4).

Maxima of monitored and predicted $PO_4$-concentrations coincide in time and have a similar size. For example, in 1973, the average $PO_4$ concentration predicted by the model was 1200 µg $PO_4$-P $L^{-1}$ and measured concentrations was on average 1300 µg $PO_4$-P $L^{-1}$. The maximal predicted $PO_4$ concentration was 2200 µg $PO_4$-P $L^{-1}$, while the maximal observed was 3000 µg $PO_4$-P $L^{-1}$. Predictions for recent years are within 15% of the observed data (e.g. 2015: Model: 133 µg $PO_4$-P $L^{-1}$, Measured 155 µg $PO_4$-P $L^{-1}$). Between 1940 and 1990, the modelled $PO_4$-concentrations show more variation. Likewise, monitored $PO_4$-data are spread more between 1967 and 1990 (Fig. 2). Before 1930, modelled $PO_4$-concentrations stabilise at levels below current observations (Fig. 4).

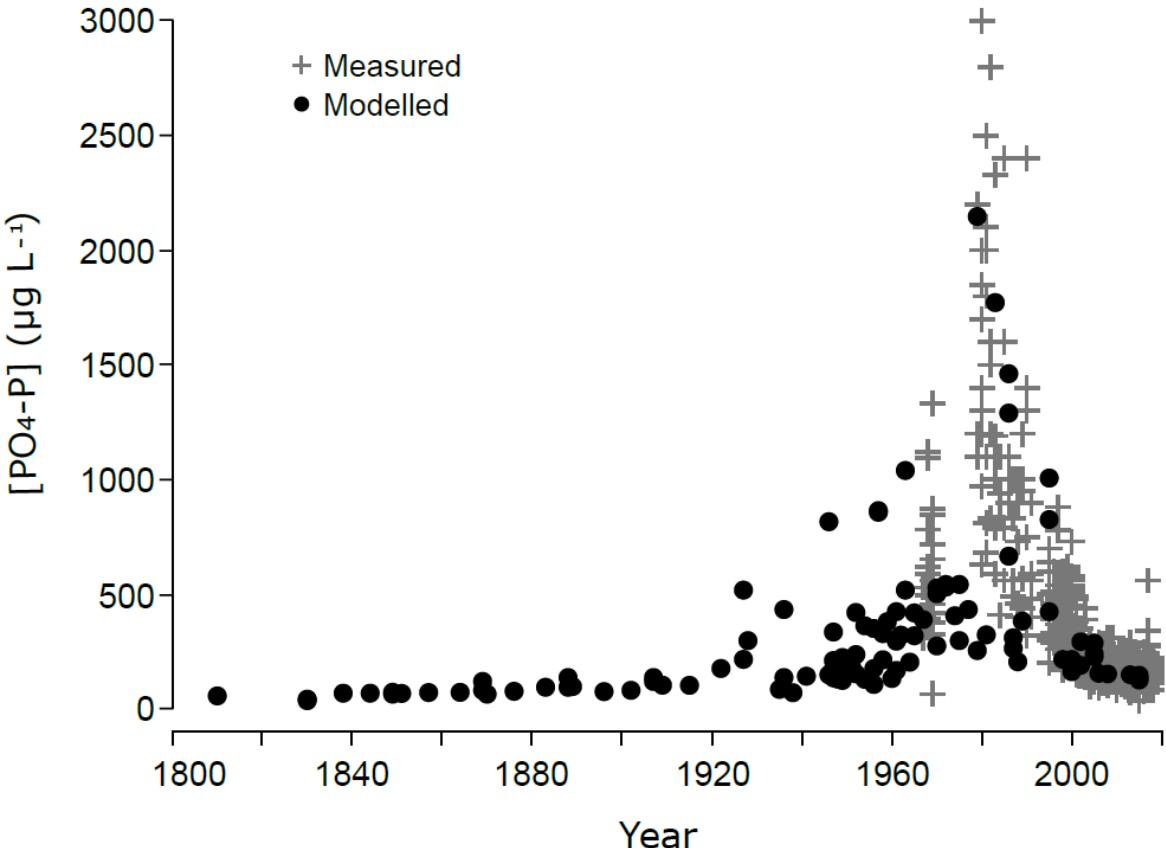

**Figure 4: Measured (grey crosses; +) and predicted (black points) of PO4-P concentrations (µgL⁻¹) in the Scheldt river in Temse. The concentrations are calculated from the sediment phosphate saturation (DPS) of the tidal marshes at Old1 and Young1, using sediment-water model 3b.**

### 3.6 Estimating background PO₄ concentrations in the Scheldt river

The deepest sediment layers are most suitable for predicting background PO₄ concentrations of the Scheldt river water. These layers are the oldest and expected to have experienced the least impact of P additions from anthropogenic sources. The Old1 marsh site was appropriate for this purpose as it developed before 1774, before the industrial revolution in Belgium. The average DPS for the bottom sediments, dated between 1800 and 1840, was 20% for core Old1 (Table 1; Fig. 2); these samples are considered to represent the pre-industrial background. That DPS value produced PO₄-concentrations of 62 µg PO₄-PL⁻¹ [95%CI (57; 66)] for the pre-industrial background, using sediment-water model 3b (Table 2). The sediment dated to 1930 had a DPS of 36%. For that DPS value, the same sediment-water model predicted a PO₄-concentration of 140 µg PO₄-P L⁻¹ [95%CI (128; 148)] (Table 2).

## 4 Discussion

### 4.1 Background vs ambient PO₄ concentrations

This work presents a novel approach to reconstruct background surface water $PO_4$ concentration in a tidal river using the DPS of adjacent tidal marsh sediments. The background concentration is essential in the context of developing local nutrient limits. The predicted pre-industrial background concentration (62 µg $PO_4$-P L$^{-1}$; Table 2) is about half of the current surface limit of the Scheldt (120 µg $PO_4$-P L$^{-1}$; Flemish Government, 1995). Remarkably, the predicted background concentrations are about a factor two larger than the background estimates of lake waters for Flanders today (15-35 µg $PO_4$-P L$^{-1}$; Cardoso et al., 2007), suggesting that the internal loading related to summer anoxia in lowland rivers contributes naturally to larger $PO_4$ concentrations in lowland rivers (see introduction). The summer peak of $PO_4$ concentration is present for five months of the year in Flanders, and can thereby have a significant influence on the mean P concentrations in the rivers (Smolders et al., 2017). Summer anoxia also occurs in eutrophic lakes, or sometimes oligotrophic brown water lakes (Nürnberg, 1995). For example, in 75 lakes in the US, anoxia occurred between zero and 83 days a year, which is less than the five months or 150 days in Flemish rivers (Nürnberg, 1995; Smolders et al., 2017).

Also, lowland rivers in Flanders are primarily groundwater-fed, as on average, 73% of streamflow can be attributed to base flow. Natural groundwater in Belgium has a median P concentration between 150 - 320 µg TP L$^{-1}$. Groundwater feeding the river waters will increase the P levels. In contrast, mostly rain fed lakes will have lower P concentrations, with rain P ranging between 1.5 and 120 µg TP L$^{-1}$ (Migon and Sandroni, 1999).

Our analysis suggests that the pre-industrial $PO_4$ concentration was about three times lower than the current concentrations in the Scheldt. For example, between 2007 and 2016, the mean $PO_4$ concentration of the Scheldt in Temse was 170 µg $PO_4$-P L$^{-1}$. However, in the 1930s, the concentration was estimated at 140 µg $PO_4$-PL$^{-1}$ and larger than current limits, at a time before widespread connection to sewer systems, P-loaded detergents, and application of mineral fertilisers.

### 4.2 Limitations of the model

Care needs to be taken with background extrapolations to ensure that post-depositional processes have not modified the biogeochemical patterns and that the area represents the area of interest (Reimann and Garrett, 2005). Several factors can obscure the reconstructed background concentrations. First, vertical migration of P can enrich deeper sediment layers, causing an overestimation of the background. Second, the sediment profiles at the tidal marshes are almost permanently saturated, and the intrusion of P-rich groundwater could affect the P concentrations in the tidal marsh sediment. Moreover, periodic flooding occurs at an approximate range of 300-350 inundations per year, depending on the tidal marsh elevation (Temmerman et al., 2003b). These conditions could favour phosphorus migration due to the reductive dissolution of Fe (oxy)hydroxides (Baken et al., 2015). We removed two cores with indications of $PO_4$ migration to address the issue (Old2 and Young2). These cores were identified by comparing the peak in the DPS age profile with the known peak in surface water $PO_4$ concentrations in the 1980s and considering the distance from the nearby creeks (Fig. 2; Fig. 1). Additionally, the DPS levels

of the deepest sediment layers were compared with layers at the surface. The surface layers had lower DPS levels than the deepest layers for one core (Young 2). The two remaining cores (Old1, Young1) had lower DPS levels in deeper sediment layers (Fig. S7). More importantly, the modelled peak in $PO_4$ concentrations based on the cores Old1, Young1 were found within two years of the monitored peak and had a similar magnitude (Fig. 4). The coinciding peaks illustrate little migration of $PO_4$ in Old1 and Young1, thereby justifying these cores as an archive for water-$PO_4$.

The limited migration is also logical: at the average DPS of 90 % in sediment showing at the peak; the sorption models predict that the solid-liquid P concentration ratio is 2900 L kg$^{-1}$, the average K value of models of Table 2. That solid-liquid ratio can be converted to dimensionless retardation factor, representing the ratio of the distance migrated by the $PO_4$ compared to the distance travelled by percolating water, of 7500 with a bulk density ($\rho_b$) of 1.3 and porosity ($\theta$) of 0.5. With a net vertical annual water percolation of about 2 meters, the retardation corresponds to a net vertical P migration rate of 2.5 cm over 100 years, i.e. vanishingly small (calculation details not shown).

Secondly, there is uncertainty on the age-depth estimation of the sampled sediment profiles. We expect that the age-depth model is most reliable for the Young1 sediment core, as it is based on a fitting of a modelled age-depth relation to four observed age-depth points, while we only had two observed age-depth points available for the other cores. Additionally, observed age-depth points were not older than 1944. Hence, the extrapolation of the age-depth model to periods before the older available age-depth points is increasingly uncertain.

### 4.3 Pre-industrial and natural background values

The population increase between 1800 and 1930 can provide a first, very crude estimate of the population-DPS relation in the Scheldt basin. In 1800 the population in Belgium was around 3 million. In 1930, this number has more than doubled to 7 million (Vanhaute, 2003). A linear relation between both suggests that the DPS is 8% for the pre-anthropogenic pristine environment, corresponding with a $PO_4$-concentration 19-41 µg $PO_4$-PL$^{-1}$, i.e. close to what researchers have indicated for pristine lakes. Such predictions need to be corroborated with older sediment observations and other archaeological information. The Scheldt river is logically more aerated than smaller lowland rivers where summer anoxia is naturally more present, i.e. the pristine PO4-P values will be higher.

### 5 Conclusions

Our study illustrated that tidal marsh sediments could estimate pre-industrial background $PO_4$-concentrations of the freshwater rivers like the Scheldt river. A sediment assessment can record and time-integrated environmental events, which provides useful spatial and temporal information. Our data estimated the pre-industrial background concentration at 62 µg $PO_4$-P L$^{-1}$ [95%CI (57; 66)], about half of the environmental limits set for surface waters in Flanders and neighbouring countries. Around 1930, the $PO_4$ levels were only about 20% lower than today, which is a remarkably large concentration at a time before the

massive application of mineral fertilisers, with lower population density and limited connection to sewer systems. The current PO$_4$ concentrations decreased by a factor ten from the peak found 40 years ago, reflecting wastewater treatment efforts and reducing diffuse P emission. It is also clear from this study that the pristine, pre-anthropogenic PO$_4$-P concentrations in the Scheldt river are well below the current ambient ones.

**Data availability**

The results of the sediment data analysis and age depth model are provided in the supplement as csv format. Results of surface water data are available upon request at the the IMIS (Flanders Marine Institute).

**Author contribution**

FL, ES and PC designed the research. FL conducted the investigation process, and developed the methodology under supervision of ES. MVDB carried out the fieldwork and conceptualised the use of the samples. ST prodived the methodology for the age-depth model and software. TM validated the use of the surface water data. EVM and FL placed the results in perspective with historical data. All the authors contributed to discussion and data interpretations, review and editing of the work.

**Competing interests**

The authors declare that they have no conflict of interest

**Acknowledgements**

This project was supported by the Research Fund Flanders (FWO), project G089319N. The results of this research greatly depended on the data collected by the OMES-monitoring and The Flemish Waterway. Many years of intensive data collection and quality assessment of the Scheldt river resulted in a unique and valuable phosphate time series. We have the utmost respect for their work and are thankful we could apply the dataset for this research. We acknowledge Dries Grauwels and Kristin Coorevits for technical assistance. We recognise the efforts from the unanimous reviewers for their constructive comments on the work, which improved the quality of the result. Finally, thanks to the Scheldt for providing this beautiful sediment archive to travel back in time and explore environmental history.

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
