# Peer review of "Phosphorus natural background estimation in the Scheldt river using tidal marsh sediment cores"

_Biogeosciences, 2021_

## Author Comment (AC2)

**Revisions manuscript bg-2021-100**

Please find below the comments of the reviewers (in black), as well as a description of how these comments were addressed in the revised manuscript (in blue).

**RC1: 'Comment on bg-2021-100', Anonymous Referee #1, 11 Jun 2021 reply**

The manuscript titled "Identification of the natural background of phosphorus in the Scheldt river using tidal marsh sediment cores" by Lauryssen et al., is a good piece of work to justify the possible sedimentary records of dissolved phosphate-P in river water in the past. Authors have analysed two types (old and young) of sediment cores from the river flood-land to establish the fact that bottom sediment retains the signature of phosphate concentration in overlying water without much alteration over the period of time. And based on those sedimentary records authors have tried to predict the "natural background" value of phosphate-P prevailed during the age of pre-industrialisation and/or any major human intervention. For this objective, one should be very careful about the possible artifacts in prediction of such background value; because that may cause unexpected errors.

*The authors would like to thank reviewer 1 for his/her kind comments and feedback and the time and energy spent reviewing the manuscript. The careful consideration will benefit the quality of this article. We appreciate the recommendations and understand the care needed in predicting a background value.*

No doubt it is an innovative attempt; but the present manuscript has some shortfalls and queries. The text language is fine but some longer statements can be avoided.

*With this reply, we solved the shortfalls and answered the questions addressed. In addition, the text was adjusted to improve flow, and clarify long statements.*

In present study it was assumed that the immobile nature of P helps in retaining water column signature in sediment. From this basic idea it is expected that the past time-zones which correspond the phosphate peaks in water should precisely match with the peaks of $P_{ox}$ or DPS found in core sediment. In Figure 2, the maximum dissolved phosphate peak appears immediately after 1980s but the most of the DPS profiles (except Young-1) in Figure 3 have peaks before 1980 (between 1920-70). In this regards some suitable explanations have to be included in the discussion section.

*We thank the reviewer for noticing the importance the identification of P preservation.*

*Section 3.3 describes the sediment core selection and represents the considerations for selecting the representative cores. Further in the manuscript, section 3.5 describes the model selection regarding the different cores to fit the model.*

*The fitting of the sorption model excluded Old2 and Young2. The DPS peaks of both cores did not correspond with what was observed in the surface water. This was indicating P migration, as was addressed in section 3.3, line 275-285.*

*The time series of Old1 displayed a DPS peak around 1960, indicating a shift of only 20 years (Figure 2).Thus the shift of Old1 is smaller than the maximal shift up to 1920 Reviewer 1 noticed. However, the core Old1 was incuded in the model fitting. That core is essential to predict the background because the core dates*

*back to 1800 at the deepest levels. Furthermore, the DPS concentrations stabilised before 1920, indicating that P has not migrated to these depths, making it suitable for background prediction.*

*Furthermore, the sorption model used to predict the surface water concentrations using observations from both cores Old1 and Young1 (model 3b, Table 2) performed better at predicting recent surface water concentrations, at relatively low concentrations. If the observations from Old1 are removed from the dataset, the background would be slightly lower with 51 µg $PO_4$-P L-1 [49; 54]. However, that model only including the observations from Young1 that underestimated recent $PO_4$ (2007-2019) values by 28%, while the model 3b had only 15% underestimation.*

*Section 3.3 was supplemented in the manuscript with part of the discussion above:*

*"The time series of Old1 displayed a DPS peak around 1960, indicating a shift of 20 years (Figure 2). However, the core Old1 was taken up for the prediction of the model fitting. That core is essential to predict the background because the core dates back to 1800 at the deepest levels. Furthermore, the DPS concentrations stabilised before 1920, indicating that P has not migrated to these depths, making it suitable for background prediction."*

Authors have estimated the level of P-saturation in sediment samples following a relation (as mentioned in the Equation 1); actually verified for the porewater-soil system of typical agricultural fields. This similar relation could be applicable to the estuarine sediment also?? Specially, I have doubt about the factor of 0.5 in this Equation 1. It may differs for sediments in aquatic environments; characterised with different geochemical settings as compared to those from agricultural lands.

*The DPS was verified for porewater-soil systems and was developed by (van der Zee et al., 1990). The 0.5 is an empirical value based on pore water measurements of non-calcareous sandy soils and is considered the sorption capacity of the soil. Lexmond et al., (1982) illustrated that the maximal sorbed P was about half the pool available after a long-term precipitation experiment. So they concluded the α was set at (0.5 ± 0.1). However, even among terrestrial soils this parameter varied between 0.3 and 0.6.*

*Because the dataset in this study had a lot of noise and a limited number of observations, we decided only to fit the parameter K, i.e. one and not two adjustable parameters. Furthermore, the two parameters α and K are correlated by fitting, if α is reduced, then K must be increased to fit the same data. The parameter α primarily affects the maximum sorption capacity, while the parameter K determines the slope between zero and the maximal capacity. The maximum is only reached at high $PO_4$ (not relevant) concentrations. We focussed on background concentrations where K is most important.*

*We can speculate that the solution-particulate P relationships are different between soils and surface water for various reasons (e.g. soil-leachates have $PO_4$ concentrations but P has equilibrated longer than in water), however we do not wat to elaborate too much to keep focus on the identiftcation on the background, all we wanted to identify is a translator between sediments and water, the validity of the "soil" model is simply the statistical validity of the predictions. In this study we chose to use the 0.5 value, but that further research is needed to asses the exact value for tidal marsh sediments.*

Ln 79-80: "In lowland rivers with tidal influence, also called estuaries such as the Scheldt estuary in Flanders, tidal marshes....." The statement is not clear; use simpler sentences.

*We used shorter sentences to clarify the statement.*

*"In Lowland rivers with tidal influence, tidal marshes develop along the river banks. tidal flooding accumulates sediment on these marshes. The process is similar to the development of river floodplains but at a much faster sedimentation rate (i.e. mm to cm per year)."*

Ln 106-07: Author has mentioned that they restrict their observation within the freshwater areas of the Scheldt estuary. Is there any specific reason for selection of only freshwater segments of that estuary??   -Here they should cite those actual reasons for such site selection in context to the aims of the present study.

*Thanks for pointing us the importance of mentioning to clarify why we restricted our research to freshwater systems.*

*The following reasons was cited in the text:*

*"The brackish portion of the estuary experiences mixing of freshwater from upland sources and seawater, making it difficult to distinguish between the anthropogenic sources of $PO_4$ from seawater influence. Furthermore, brackish and saltwaters in the North Sea have $PO_4$ concentrations about a factor 10 lower compared to freshwater (Burson et al., 2016). Too low concentrations could lead to an undetectable signal in the sediment. Our research was focussed on freshwater and lowland river systems, and the human influence on the $PO_4$ concentrations. "*

Ln 120-121: Authors should mention about the site specific sedimentation rates (as obtained from radiometric dates) at four sampling locations. As the sediment accretion rates in this estuarine system are quite variable (e.g. 0.3-3.2 cm/yr); this information would provide better perception regarding the 'young" and "old" settings.

*We agree with reviewer 1 that comparing specific sedimentation rates associated with young and old marshes gives a better perception of the setting. These data were collected from literature and will be added to the text:*

*"For example, between 1931 and 1951 young marshes accumulated at rates of 1.58 to 3.22 cm $yr^{-1}$, during the subsequent period 1955-2002 they accumulated at lower rates of 0.4-1.84 $cm^{-1} yr^{-1}$. In contrast to the young marshes, the elevation of old marshes is at any time very close to MHWL of 0.32 to 0.58 cm $yr^{-1}$ in the Western Scheldt (Temmerman et al., 2003)."*

 Ln 150-155: This section of sediment analyses needs more details. It has "…..solid-liquid ratio of 1 g in 50 ml….". What is that solid-liquid ?? Here, it looks solid-liquid refers sediment-oxalate solution. Provide the details about the liquid and its purity, concentration etc used for this leaching experiment.

*We understand reviewer 1, and therefore will clarify the purity of the liquids and concentrations used for the oxalate extraction.*

*"The extractant was made with Milli-Q® grade water and all glassware used was acid washed prior to remove all P. That acid oxalate extractant, a mixture of ammonium oxalate and oxalic acid at pH = 3, targets poorly crystalline oxyhydroxides of Fe, Al and Mn and the associated P.*

*…*
*The preparation of extraction solution and dilutions were made with ultrapure water; all glassware used was acid-soaked overnight in a 1% HCl acid bath. The extraction solution is composed of a mixture of 0.2 M ammonium oxalate and 0.2 M Oxalic acid at pH 3, as described by Schwerzman (1964). The extraction was done with 1 g of dry sediment in 50 ml of extraction solution."*

Ln 250: In the Table 1, authors should provide the estimated $Fe_{ox}$ and $Al_{ox}$ values for each samples. This would help reader to verify the DPS values presented in the table.

*The addition of $Fe_{ox}$ and $Al_{ox}$ concentrations to table 1 allows the reader to calculate and verify the DPS values presented in the text. We agree with the reviewer that this will benefit the interpretation of the value. The concentrations will be added to the final table.*

Ln 295: In Table 2, there are two background values (for 1800 and 1930) have been presented. Its bit confusing; which one represents the natural background value. Is 1930s value is really essential??

*We thank reviewer for pointing out that confusion, we kept 1800 defined as pre-industrial background but did not use any longer the term background for the 1930 estimate, we merely defined it as the concentrations around 1930. The concentrations near 1930 is used because it refers to a time before the widespread application of mineral fertilisers, industrialisation and sewage connection. Therefore $PO_4$ concentrations were considered to be lower than today. However, the authors were surprised that the values presented in 1930 were already at levels similar to today.*

Secondly, in the Table 2 caption, it has mentioned that the profiles of only one site (Old-1) has used for prediction of background phosphate concentration. However, in same table, the model 3b is based on Young-1 and Old-1 data set). This is also confusing.

*We thank reviewer 1 for mentioning another confusing statement in table 2. The only core dating back to pre-industrial times (1800) was Old1, with a DPS of 20% at the bottom of the core. However, model 3b converted DPS to surface water $PO_4$ was fitted on both Old1 and young1 core data. The data of core Young1 is valuable for fitting the model but does not go back in time long enough as it wasy formed after 1945. So, the model fitted on both cores (Old1 + Young1) was applied to the deeper layers of the Old1 core, to predict the background.*

*Caption change*

*"Table 2: The predicted concentrations of phosphate (PO4-P µgL-1) in the Scheldt river based on the Degree of Phosphate Saturation (DPS) in the sediment layers of marsh Old1, dating back to 1800 (pre-industrial), where DPS values stabilised with depth at 20% , and the predicted concentration dated to 1930 where DPS stabilised at 36%. The Pbias measures the average tendency of the simulated data to be larger or smaller than their observed counterparts, expressed as a percentage (Moriasi et al., 1983). Conversion of DPS to river phosphate concentration based on the association of DPS with $PO_4$-P calibrated to data 1967-2016, thereby using different calibration for sediment-water models; the details of models are in Table S2. Model 3b (in bold), is proposed as the most accurate one (see text).*

Ln 309-310: The models using mean DPS and individual PO4 are considered more useful for evaluating the factor "K". Here authors should mention the proper reasons for this choice.

*We use DPS (in X-axis) to predict solution P (Y-axis). Both X and Y are prone to sampling error, hence we are investigating allometric relationships. Allometry is not commonly used and certainly not in non-linear allometric relationships. Hence, we reduced it to regression and try to limit error in the X-axis, the assumed independent variable. To limit that variation, we averaged data. A sediment analysis gives an average DPS signal over a deposition period. However, due to variation in the marsh surface, the age-depth relation can vary slightly. By taking an average DPS concentration, we compensate for this natural variation. The mean DPS over different replicate cores, therefore, corresponded to each surface $PO_4$ measurement.Moreover, as we had described from line 311, single DPS value requires the need to duplicate/ triplicate solution P data (e.g. three DPS values for one year of water P data), which incorrectly increases the number of observations in the dataset.*

*Part of this reasoning was already described in lines 311-313 of the reviewed manuscript.*

*The following statement will be added to the text:*

*"A single sediment sample analysis gives an average P signal over a deposition period. However, due to variation in the marsh surface, the age-depth relation can vary slightly. By taking an average DPS from replicate cores, we reduced the the variation in the independent variable with which the dependent variable ($PO_4$ in water) is predicted. Furthermore, in most models, the prediction error increased by relating individual rather than mean DPS values with individual $PO_4$ measurements."*

Ln 356-357: The summer anoxia induced oxide dissolution has projected for excess P loads in river water and thus explained for higher projected background value of 62 microgm/l. Summer anoxia is generally short-term seasonal process and therefore is it really could be effective to maintain higher background over the long run?? Or, it is some other processes responsible for such projected higher values in river water relative pristine lakes. Similar effects of seasonal anoxia are not evident in those lake waters??

*We thank the reviewer for questioning the effect of summer anoxia on the background concentrations. Hereby we clarify the importance of this process in lowland rivers.*

*The summer peak of $PO_4$ is present for five months of the year in Flanders, thereby significantly influencing the mean P concentrations in the rivers (Smolders et al., 2017). Therefore, we expect the summer anoxia to be the main driver behind the elevated P concentrations.*

*Also, lowland rivers in Flanders are primarily groundwater-fed; on average, 73% of streamflow can be attributed to base flow. Natural groundwaters in Belgium have a median P concentration between 150 - 320 µg P $L^{-1}$. Groundwater feeding river waters will increase P levels. Conversely, primarily rain-fed lakes will have lower P concentrations, ranging between 1.5 and 120 µg $L^{-1}$.*

*Summer anoxia also occurs in eutrophic lakes, or sometimes oligotrophic brown water lakes (Nürnberg, 1995). For 75 in the US lakes anoxia occurred between zero and 83 days a year, less than the five months or 150 days in Flemish rivers (Nürnberg, 1995; Smolders et al., 2017).*

*The reasoning presented above will be added to the manuscript.*

Ln 400: "Our data estimated that the pre-industrial background concentration is about half of the….." In this conclusionary statement, mention the actual predicted background value i.e., 62microgm PO4-P/l along with confi. Intervals.

*Reviewer 1 brought up a good point to add the numerical value and confidence limits to the conclusion statement; we agree that it is crucial to mention there.*

*The sentence at line 400 will be changed to: "Our data estimated that the pre-industrial background concentration is 62 µg PO$_4$-P [95% CI (57; 66)] which is about half of the environmental limits set for surface waters in Flanders and neighbouring countries. "*

The plots of Figure S9 are easy to visualise as compared to those in Figure S7 and S8. Therefore, S7 and S8 can be presented in similar format of S9. Furthermore, in these depth profiles, the extremes (Highs and lows) should marked with corresponding ages. In my opinion instead of supplementary documents, these depth profiles would be good to present as the part of the main text.

*We thank reviewer 1 for proposing a more convenient way to visualise figures S7 and S8. The orientation of S7 and S8 has changed accordingly.*

[Figure]

*Figure S7: Depth profiles of the Degree of P-saturation (%)*

*However, we decided not to add the figure to the main text as it contains very similar information as Figure 3, presenting the DPS timeline calculated based on the depth profiles from Figure S7. The peaks and corresponding dates are presented in Figure 3.*

**RC2**: 'Comment on bg-2021-100', Anonymous Referee #2, 30 Aug 2021 reply

The paper does address relevant and appropriate scientific questions which are very important to environmental P cycling and the journal. The authors propose to use sediment cores as a chemical archive to estimate historical surface concentrations of orthophosphate (orthoP) using a Langmuir-type sorption model in order to test statements regarding orthoP pre-industrial levels in freshwater rivers.

*We thank reviewer 2 for noticing the value and importance of our work. Furthermore, we highly appreciate the effort put into the review and formulation of constructive comments.*

The manuscript is relevant and valuable work, but the manuscript as submitted needs significant work with regard to the Introduction and language throughout. There are many areas where the language struggles and the meaning of the sentence is lost. I would recommend authors seek editing advice for English language, as many of the difficulties maybe due to language barriers.

*The manuscript was revised on language, and complex sentences were adjusted to clarify their meaning.*

Because the paper uses a model to predict pre-industrial orthoP concentrations the authors should be careful to use language to that effect throughout the document. Language such as estimates, predictions, suggestions rather than words like identification (title). The name of the model should be mentioned in the abstract - after reading the abstract I had no idea a model was being used as the main methodology.

*We thank reviewer 2 for pointing out the importance of the language used to describe background predictions. We understand the importance of language in this context.*

*Therefore we adjusted the title to: "Phosphorus natural background estimation in the Scheldt river using tidal marsh sediment cores."*

*The specific model used in the manuscript was not described in the abstract.*

*Therefore the following sentence was added at line 21:*

*"By combining the sediment-P and water-$PO_4$ data, the sorption characteristics of the sediment could be described using a Langmuir-type sorption model."*

The Introduction needs to be rewritten with current literature on global P problems in the worlds waterways and why estuaries/lowland rivers need to be studied to determine P capacity and/or leakage. I would recommend that the Introduction be rewritten to include some of the literature referenced in the methods section. I had a much better idea about what the authors were investigating after reading the methods section – and that should be reflected in the Introduction. For example, the Introduction should introduce why P is a problem globally, and why estuaries/lowland rivers are particularly a problem for both P storage and leakage – although not enough is known about either problem in these areas. There also needs to be a thorough discussion of why P in sediments would reflect overlying orthoP concentrations – leading the reader as to why you chose to use the methods described. And finally, a thorough discussion of the specific P problems related to this area of Belgium. After reading the Abstract and Introduction compared to the Methods section I had very different ideas about what was going to be discussed in the paper.

*We thank reviewer 2 for the valuable suggestion to improve the Introduction. The idea about the investigation was clarified, and the Abstract – Introduction and methods were alligned.*

*The Introduction was rewritten according to the suggestions discussed, the latest literature was consulted.*

*The following structure was followed:*

- *The global P problem, importance to limit P concentrations*
- *EU Water framework directive, maximum P concentrations are defined.*
- *Importance of lowland regions, P holding capacity and leakage?*
  - *Excessive P levels are especially a problem in lowland rivers as they exceed However, the natural background for P in rivers is largely unknown. These background values are important to develop nutrient limits.*
- *Correlation P sorption & Pore water concentrations (including Literature from Methods section)*
  - *The relation between soil characteristics and pore water PO4 concentrations have been described by several authors in the Belgium and the Netherlands. However the relation between sediments and surface waters has not yet been investigated.*
- *Why is P in sediment reflecting overlying ortho P concentrations (DPS relation)*
- *Importance of Sediment analysis for Long term reconstruction of P loading*
- *Measuring P using new method (tidal marsh sediment cores)*
  - *Specifically Tidal marshes along the Scheldt have been investigated regarding tidal marsh growth and carbon sequestration, however P storage analysis are lacking.*
  - *Therefore we will analyse the P accumulation in tidal marsh sediment cores to combine knowledge from both soil sorption models and tidal marsh elevation models. The goal is then to estimate the pre-industrial background to serve as method to develop regionally specific nutrient limits for lowland rivers.*

I will try and identify specific examples regarding my overall comments above as well as highlight the great parts of the manuscript.

*We appreciate the addressing the specific examples regarding language statements and the general appreciation of the manuscript.*

Recommend defining P forms early in the manuscript (Introduction) and sticking to that nomenclature throughout the document. For example, orthophosphate for dissolved P or reactive P (Line 38), Total P as organic, and inorganic – all forms of P. I would also recommend using the nomenclature $L^{-1}$ rather than /L (Line 41) - unless that is mandated by the journal.

*The author agrees that one nomenclature should be used consistently throughout the document, we chose for ortho phosphate ($PO_4$).*

*The nomenclature $L^{-1}$ was changed throughout the document (Corrected)*

The sentence on line 38 will be replaced by:

*"As different P forms occur in surface waters, we focussed on ortho-phosphate ($PO_4$), which is almost identical to the reactive P determined by a colour reaction. Other P forms present in surface water include organic P fractions and P sorbed to mineral colloids. Total P refers to all P forms together. "*

The use of historical orthoP data from collections using DPS sediment data from adjacent marshes is a novel concept of how to predict pre-industrial level orthoP concentrations – and will help with predictions on P loading for the future in these types of environments.

We thank the author for addressing the relevance of our research.

Recommend a sentence in the abstract from Discussion 4.2 Limitations of the model such as careful consideration for P-migrations, checks or correlations on DPS and orthoP peaks when recreating orthoP levels or history in areas prone to excess orthoP or eutrophication.

The following sentence was added to the abstract:

*"The model requires a careful consideration of vertical P migration and correlates sediment DPS and water $PO_4$ to reconstruct historical concentrations"*

Specific examples of rough or misguided or language barriers includes Line 181, Line 215, Line 324-325, Line 357.

Line 181: "Therefore, we rescaled the modelled marsh surface elevation by fitting it through the observed depth-age points from the GPS measured marsh surface elevation in 2016 and previously published radiometric and paleoenvironmental dating (Temmerman et al. 2003; 2004)."

*The statement was reformulated in the final manuscript:*

*Because this study focussed on the time of sediment deposition, the original age-depth relation calculated by MARSED was recalibrated by using the observed age-depth point. The observed age-depth points originate from GPS measured marsh elevation in 2016 (Van de Broek, unpublished data) and previously published radiometric and paleoenvironmental dating and act as ground truth (Temmerman et al. 2003; 2004).*

Line 215: We did not use the existing sorption models for soils directly, and instead, the parameter K, in Eq. (2), was calibrated to the most recent Scheldt water data. So that, the K-value adapted to the geochemistry of the tidal marsh sediments.

*The statement was reformulated in the final manuscript*

*As the parameters (K – L $kg^{-1}$) of existing soil models had been calibrated for soil - pore water concentrations, the sediment-water relation is unlikely equal. Therefore, the model was calibrated by fitting parameter K on sediment DPS measurements and recent Scheldt water $PO_4$ measurements. As a result, the fitted parameter K is adapted to the local chemistry of tidal marsh sediments and of the surface water.*

Line 324-325: In contrast, half of the observations were underestimated by at least 25% for model 2b and by 11% for model 3b (Table 2). The actual by predicted plots illustrate the same (Fig. S10). Based on these observations, model 3b was considered as the best model, although the residual standard error (RSE) was lower for model 2b (Table S2). Model 3b predicted recent PO4-concentrations best, with median underestimation of only 11% (Table 2). The selected model 3b succeeds to reconstruct the rise and fall in surface water PO4-concentrations based on the sediment characteristic DPS (Fig. 4).

*We thank the reviewer for pointing out that the paragraph about the model performance was hard to read. Therefore, we rewrote the passage and included the Pbias as a model evaluation.*

*Section Evaluation Model Performance:*

*"Additionally, the Percent bias (Pbias) was calculated for data points between 2007 and 2016. The Pbias measures the average tendency of the simulated data to be larger or smaller than their observed counterparts, expressed as a percentage (Moriasi et al., 1983; Eq. 3)."*

*Line 324-325 will we replaced by the following paragraph:*

*"Model 3b was considered the most suitable for predicting background concentrations. The Pbias was the lowest for recent observations of model 3b, as simulated data was underestimating the observations by -14.9 %, which is within the acceptable range of ±25% (Moriasi et al., 1983). However, model 2b had an average underestimation of more than 28% and model 1b overestimated recent observations by more than 60%, which is unwanted for calculating the background, and therefore both model fits were considered unsuitable (Table 2). The actual by predicted plots illustrate the same (Fig. S10). Based on these observations, model 3b was considered as the best model, although the residual standard error (RSE) was lower for model 2b (Table S2). The selected model 3b successfully reconstructed the rise and fall in surface water PO4-concentrations based on the sediment characteristic DPS (Fig. 4)."*

The methods section is well written. The language in the methods with regard to calculations of P-saturation and P sorption capacity is great – it would be great to see some of this language put into the Introduction as to why measuring P this way is important in gauging effects of long-term P loading – correlations between P sorption and pore water concentrations. The authors also used a published Langmuir-type sorption model to predict orthoP concentrations that had been used before for sediments and water in this area – which makes it very relevant in this study.

The authors appreciate the positive comments on the methods section. The literature on P saturation was partly taken up in the Introduction to bring a complete story.

Two topics will be discussed in the Introduction as well, as was mentioned in the fourth comment of reviewer 2.

- *Correlation P sorption & Pore water concentrations (Literature Methods)*
- *Why is P in sediment reflecting overlying ortho P concentrations (DPS relation)*

Expected a discussion of why previously reported/estimated values for background orthoP in this lowland river is larger than estimates for background P in other lakes/rivers (15-35 ug P/L) what might be causing excess surface orthoP concentrations in these lowland areas pre-industrial era.

*The comment was already addressed in the reply to the comment of reviewer 1 about Ln 356-357*

- The specific conditions in lowland rivers – slow flow, summer anoxia in small rivers is present for 5 months a year
- Rivers are largely fed by baseflow (75%) bringing in a significant source of P. The average groundwater P concentrations are 145-325 µg/L
- Lakes fed by rainwater will have lower P concentrations at rainwater P varies between (*1.5 and 120 µg L$^{-1}$*)

The results from sediment core selection – the discussion on P-migration for the two samples (Old2 and Young2) and why they should be eliminated from the analysis of DPS and PO4 relationship is appropriate, understandable, and very relevant.

*The elimination of two cores showing issues with P preservations are indeed important for the results. Thanks for pointing this out. We also mentioned the importance of core selection in the abstract.*

**References**

Burson, A., Stomp, M., Akil, L., Brussaard, C. P. D. and Huisman, J.: Unbalanced reduction of nutrient loads has created an offshore gradient from phosphorus to nitrogen limitation in the North Sea, Limnol. Oceanogr., 61(3), 869–888, doi:10.1002/LNO.10257, 2016.

Lexmond, T. M., Riemsdijk, W. H. van and Haan, F. A. M. de: Onderzoek naar fosfaat en koper in de bodem in het bijzonder in gebieden met intensieve veehouderij, L.H. [online] Available from: https://research.wur.nl/en/publications/onderzoek-naar-fosfaat-en-koper-in-de-bodem-in-het-bijzonder-in-g (Accessed 15 September 2021), 1982.

Moriasi, D. N., Arnold, J. G., Liew, M. W. Van, Bingner, R. L., Harmel, R. D. and Veith, T. L.: MODEL EVALUATION GUIDELINES FOR SYSTEMATIC QUANTIFICATION OF ACCURACY IN WATERSHED SIMULATIONS, Trans. ASABE, 50(3), 1983.

Nürnberg, G. K.: Quantifying anoxia in lakes, Limnol. Oceanogr., 40(6), 1100–1111, doi:10.4319/LO.1995.40.6.1100, 1995.

Smolders, E., Baetens, E., Verbeeck, M., Nawara, S., Diels, J., Verdievel, M., Peeters, B., De Cooman, W. and Baken, S.: Internal Loading and Redox Cycling of Sediment Iron Explain Reactive Phosphorus Concentrations in Lowland Rivers, Environ. Sci. Technol., 51(5), doi:10.1021/acs.est.6b04337, 2017.

Temmerman, S., Govers, G., Meire, P. and Wartel, S.: Modelling long-term tidal marsh growth under changing tidal conditions and suspended sediment concentrations, Scheldt estuary, Belgium, Mar. Geol., doi:10.1016/S0025-3227(02)00642-4, 2003.

van der Zee, S. E. A. T. M., van Riemsdijk, W. H. and de Haan, F. A. M.: HET PROTOKOL FOSFAATVERZADIGDE GRONDEN., 1990.

---

## Author Response (AR2)

**Revisions manuscript bg-2021-100**

Please find below the comments of the reviewers (in black), as well as a description of how these comments were addressed in the revised manuscript (in blue).

**Anonymous Referee #1 : nominated 29 Nov 2021, accepted 04 Dec 2021, report 16 Dec 2021**

**Report #1**

I went through the revised MS carefully and according to me, the revised draft of the MS has improved a lot in terms of many scientific and technical aspects; however, authors should think about few minor issues before finalizing it.

In the first draft, the language used in the text was a major concern (another reviewer has also mentioned about the same). For this revised MS also, the polishing of language is still very essential to bring more clarity in the text. For example, the modified title of the MS itself (ie., Phosphorus natural background estimation in the Scheldt river using tidal marsh sediment cores) is not very clear and it is better to modify it. It would sound better if it is simply rearranged as: "Evaluation of natural background of phosphorus in the ……………". Similarly, in the main text also the meanings of some statements are not very clear and for common reader it would be difficult to get the actual concept of the topic. Author should go for simple sentences; instead of many complex or compound statements. For such corrections /changes in the main text, a native English speaker could provide necessary helps. In the text there are number of abbreviated terms has been used without actual full forms (e.g. IMIS, OMES, MARSED etc). Provide the extended forms of each of those abbreviated terms at least once (probably at the first appearance).

Ln 35: "Eutrophication by excess nutrients, including P and nitrogen (N), can lead to……" In this sentence instead of P and N; use phosphate and nitrate/nitrite as nutrients. Ln 62-63: Before that, large scale agriculture dates back to the middle ages and the Roman period. …. This sentence looks incomplete. Change it to bring out the actual meaning.

The authors thank the reviewer for acknowledging the improvement of the article after the revisions and pointing out the language concerns.

The manuscript's language was again revised by two persons , who are experienced with writing English academic publications. In addition, the introduction was shortened by 15% to improve the text's readability and flow and coherence.

Complex and compound statements have been adjusted and clarified in the main text. This was done by simplifying longer statements, and using a writing assistant. Overall, the average number of words per sentence is 19 situated between 18 and 25 words. Paragraphs contain about nine sentences within the average range for academic texts.

The title was adjusted according to the suggestions of the reviewer: "Estimation of Phosphorus natural background estimation in the Scheldt river using tidal marsh sediment cores"

The abbreviations were described in full forms. IMIS was described in full at line 168. The full description of OMES was added to line 170: "Research program environmental effects Sigmaplan".

The abbreviation MARSED was described in full form at line 220: "time-stepping marsh sedimentation model (MARSED)"

In line 32 reference to the nutrients was adjusted as phosphate ($PO_4$), however, the authors considered that referring to nitrogen (N) covers the whole load as nitrate, nitrite, ammonium and organic components add to the nutrient.

In line 62 the sentence was not taken up in the final manuscript after editing the introduction.